# Macrophage Inhibitor Clodronate Enhances Liver Transduction of Lentiviral but Not Adeno-Associated Viral Vectors or mRNA Lipid Nanoparticles in Neonatal and Juvenile Mice

**DOI:** 10.3390/cells13231979

**Published:** 2024-11-29

**Authors:** Loukia Touramanidou, Sonam Gurung, Claudiu A. Cozmescu, Dany Perocheau, Dale Moulding, Patrick F. Finn, Andrea Frassetto, Simon N. Waddington, Paul Gissen, Julien Baruteau

**Affiliations:** 1Great Ormond Street Institute of Child Health, University College London, London WC1E 1EH, UK; loukia.touramanidou.17@ucl.ac.uk (L.T.); sonam.gurung@ucl.ac.uk (S.G.); claudiu.cozmescu.15@ucl.ac.uk (C.A.C.); d.perocheau@ucl.ac.uk (D.P.); d.moulding@ucl.ac.uk (D.M.); p.gissen@ucl.ac.uk (P.G.); 2Moderna, Inc., Cambridge, MA 02139, USA; patrick.finn@modernatx.com (P.F.F.); andrea.frassetto@modernatx.com (A.F.); 3Institute for Women’s Health, University College London, London WC1E 6HX, UK; s.waddington@ucl.ac.uk; 4Wits/SAMRC Antiviral Gene Therapy Research Unit, Faculty of Health Sciences, University of Witswatersrand, Johannesburg 2193, South Africa; 5Great Ormond Street Hospital for Children NHS Foundation Trust, London WC1N 3JH, UK

**Keywords:** gene therapy, AAV, LNP mRNA, lentiviral vector, LV, macrophages, macrophage inhibitors, liver, transduction, spleen, hepatocytes, expression

## Abstract

Recently approved adeno-associated viral (AAV) vectors for liver monogenic diseases haemophilia A and B are exemplifying the success of liver-directed viral gene therapy. In parallel, additional gene therapy strategies are rapidly emerging to overcome some inherent AAV limitations, such as the non-persistence of the episomal transgene in the rapidly growing liver and immune response. Viral integrating vectors such as in vivo lentiviral gene therapy and non-viral vectors such as lipid nanoparticles encapsulating mRNA (LNP-mRNA) are rapidly being developed, currently at the preclinical and clinical stages, respectively. Macrophages are the first effector cells of the innate immune response triggered by gene therapy vectors. Macrophage uptake and activation following administration of viral gene therapy and LNP have been reported. In this study, we assessed the biodistribution of AAV, lentiviral, and LNP-mRNA gene therapy following the depletion of tissue macrophages by clodronate pre-treatment in neonatal and juvenile mice. Both neonatal and adult clodronate-treated mice showed a significant increase in lentiviral-transduced hepatocytes. In contrast, clodronate pre-treatment did not modify hepatocyte transduction mediated by hepatotropic AAV8 but reduced LNP-mRNA transfection in neonatal and juvenile animals. These results highlight the importance of age-specific responses in the liver and will have translational applications for gene therapy programs.

## 1. Introduction

Over the last two decades, gene therapy has transformed the therapeutic landscape of liver monogenic diseases demonstrating maturity with numerous clinical successes [1,2,3]. Adeno-associated viral (AAV) vectors have emerged as a leading strategy in liver-targeting gene therapy [4,5,6,7,8]. However, systemic administration of high doses of AAV vectors have shown limitations caused by severe innate and adaptive immune responses [9,10,11,12], preventing re-injections in humans [13]. AAV vectors deliver mainly episomal transgenes, which are not passed to daughter cells during cell division and liver growth [14,15,16]. Therefore, sustained clinical efficacy in a rapidly growing paediatric liver requires alternative gene therapy strategies such as integrative approaches, e.g., in vivo lentiviral vectors [17,18], gene integration mediated by nucleases [19,20] or not [21], or non-viral technologies, e.g., lipid nanoparticles (LNP) encapsulating mRNA (LNP-mRNA) [22,23,24,25,26,27].

Whatever the chosen gene therapy strategy, methods to optimise hepatocyte transduction are essential for efficacy, safety, and cost-effectiveness. Administering a minimal effective dose improves the safety profile as some viral vectors have shown dose-dependent severity of adverse events [28]. Macrophages are the first effector cells for innate immunity. As such, liver-targeting lentiviral gene therapy in vivo has shown high uptake by macrophages in the splenic marginal zone reducing efficacy [17,18,29,30]. Macrophage activation following AAV vector administration has been reported [31]. Similarly, LNPs can trigger innate immunity by uptake from antigen-presenting cells [32].

Clodronate (dichloroethylene-bisphosphonate or Cl2MBP) is a bisphosphonate molecule with market authorisation in cancer. Clodronate is a hydrophilic molecule that can be entrapped between concentric phospholipid bilayers to form artificial spheres or liposomes [33]. Liposome-encapsulated clodronate is preferentially uptaken by macrophages [34]. Following the degradation of phospholipid bilayers by lysosomal phospholipases, clodronate is metabolised intracellularly to cytotoxic adenosine triphosphate (ATP) analogue, β,γ-Dichloromethylene ATP, leading to macrophage apoptosis [35] (Appendix A). Clodronate-encapsulated liposomes achieve a transient depletion of circa 90% macrophages in both red pulp of the spleen and Kupffer cells in the liver at 24 h after systemic injection [34,36]. The transient depletion of macrophages lasts for 1–2 weeks [37]. Adenoviral and AAV vectors result in the activation of the innate immune system leading to the elimination of transduced cells [38,39,40,41,42,43,44,45]. The resident liver and the splenic macrophages act as triggers of the initial non-specific immune response against pathogens and are accountable for most absorbed vector particles [17,29,30,46,47,48,49]. Pre-administration of clodronate liposomes followed by administration of adenoviral vectors depleted macrophages and allowed higher liver transduction in vivo [50,51,52]. In contrast, pre-administration of clodronate and AAV vector injection in vivo produced a considerable reduction in transgene expression in the liver [53].

In this study, we investigated the effect of clodronate-mediated macrophage depletion on liver transduction in neonatal and juvenile mice. Specifically, we depleted liver and spleen macrophages prior to administering three gene therapy modalities: lentiviral vector, AAV vector, and non-viral LNP-mRNA. Our results show that macrophage depletion induced by systemic administration of clodronate liposomes enhances hepatocyte transduction by lentiviral vectors but does not benefit AAV vectors or LNP-mRNA.

## 2. Materials and Methods

### 2.1. Experimental Design

The efficacy of intravenous infusions was assessed after 4 weeks following a single viral vector infusion for both lentiviral and AAV vectors and 24 h after LNP.mRNA infusion. Four weeks of age represents an age when most of the mouse liver growth has been achieved [54]. The transgenic expression following LNP.mRNA is limited to a few days [55] and a timeline of 24 h was chosen to capture a peak of GFP expression.

### 2.2. Vector Production and Formulation

VSV.G-pseudotyped third-generation self-inactivating (SIN) lentiviral vectors carrying the Green fluorescent protein (GFP) transgene were produced by transient transfection into HEK293T cells. Producer cells were transfected with a solution containing the selected lentiviral vector transgene backbone, the packaging plasmids pMDLg/pRRE and pCMV.REV, pMD2.G (Plasmid Factory, Bielefeld, Germany) and polyethylenimine (PEI) (24765, Polysciences, Warrington, PA, USA. Transfection media was changed after 4 h, and the supernatant was collected 48 h after the media change. Lentiviral vector-enriched supernatant was then sterilised through a 0.22 μm filter and ultracentrifuged at 23,000 rpm for 2 h. The pellet containing the vector particles was then resuspended in small volumes of phosphate buffer saline (PBS), aliquoted, and stored in −80 °C. After virus collection, a titration step was performed by transduction of HEK293T cells with the lentiviral vector at different dilutions. Seven days later, the transduced cells were collected, and qPCR was performed for quantification of vector genomes per mL. The AAV vector was provided by Vector Biosystems Inc. (Malvern, PA, USA). The vector was produced by triple transfection in HEK293T cells and was purified through a series of cesium chloride centrifugations. The transgene plasmid presented the following sequence: a GFP transgene under the transcriptional activity of the LP1 promoter and with the Woodchuck Post-Regulatory Element (WPRE) downstream of the transgene. 

*GFP* encoding mRNA encapsulated in Lipid Nanoparticles (LNPs) were provided by Moderna Inc. (Cambridge, MA, USA) using their proprietary technology. *GFP* mRNA was synthesised in vitro by T7 RNA polymerase-mediated transcription, initiated with a cap, followed by a 5′ untranslated region (UTR), an open reading frame (ORF) encoding *GFP*, a 3′ UTR and a polyadenylated tail. Uridine was globally replaced with N1-methylpseudouridine. For in vivo intravenous delivery, LNP formulations were generated. Briefly, mRNA was mixed with lipids at a molar ratio of 3:1 (mRNA:lipid), as previously described [55]. mRNA-loaded nanoparticles were exchanged into a final storage buffer and had particle sizes of 80–100 nm, >80% encapsulation of the mRNA by RiboGreen assay, and <10 EU/mL endotoxin concentrations.

### 2.3. Animals

Animal procedures were approved by institutional ethical review and performed per UK home office licenses PP9223137, compliant with ARRIVE and NC3R guidelines. Wild-type C57BL/6 and CD1 mice were purchased by Charles River (Harlow, UK) and maintained on standard rodent chow (Harlan 2018, Teklab Diets, Madison, WI, USA) with free access to water in a 12 h light/12 h dark environment. Clodronate and PBS liposome were purchased by Stratech (Ely, UK) (references F70101C-N-FOR and F70101-NL-FOR). Systemic administration in both neonatal and 2.5-week-old animals was performed by repeated intraperitoneal injections at 6 and 30 h prior to intravenous injection of viral or non-viral gene therapy vectors as per manufacturer’s dose recommendation (0.12 g/kg). Vector administration was carried out by intravenous superficial temporal vein or tail vein injection for neonatal and 2.5-week-old mice, respectively. The lentiviral vector dose was 4e10TU/kg for all treated animals while 1e13Vg/kg and 1 mg/kg were the dose for AAV vector and LNP-mRNA injections, respectively. Lentiviral and AAV vector-injected mice were harvested at 4 weeks following vector injection while LNP-mRNA-injected animals were harvested at 24 h post vector injection.

### 2.4. Vector Copy Number

Following liver perfusion, liver and spleen samples from lentiviral and AAV vector-injected mice were rapidly frozen using dry ice and stored at −80 °C until genomic DNA extraction. The QIAgen DNeasy Blood & Tissue Kit (69504, QIAgen, Hilden, Germany) was used for genomic DNA extraction, following the manufacturer’s guidelines. The plasmid standard curve was prepared by serial dilutions ranging from 10^7^ copies/5 μL to 10^3^ copies/5 μL of a plasmid, containing titin, and WPRE sequences. For lentiviral vector genome copies, the WPRE sequence was used with the following set of primers 5′-TGGATTCTGCGCGGGA-3′ (forward), 5′-GAAGGAAGGTCCGCTGGATT-3′ (reverse), 5′-FAMCTTCTGCTACGTCCCTTCGGCCCT-TAMRA-3′ (probe). The housekeeping gene *titin* was used for quantification of the cell was used to normalise the results, using the following primers; for titin: 5′-AAAACGAGCAGTGACGTGAGC-3′ (forward), 5′-TTCAGTCATGCTGCTAGCGC-3′ (reverse), 5′-56-FAM/TGCACGGAAGCGTCTCGTCTCAGTC/3BHQ_1-3′ (probe). TaqMan Universal PCR Master Mix (4304437, Thermo Fisher, Dartford, UK) was used to amplify the region of interest. The standard cycling conditions were used, starting with an initial step at 50 °C for 2 min, followed by a 10 min activation step at 95 °C, and then 40 cycles of denaturation at 95 °C for 15 s, annealing primers at 72 °C for 1 min, and extension at 60 °C for 1 min in a qPCR machine (4376357, Thermo Fisher, Dartford, UK).

### 2.5. Immunohistochemical Staining

At harvest, liver and spleen samples were fixed in 10% formalin solution, left at room temperature for 48 h before transfer, and stored in 70% ethanol at 4 °C. The liver was paraffin-embedded and sectioned at 5µM thickness. The resulting slides were then kept at room temperature until staining. Sections were dewaxed in Histoclear (NAT1330, Scientific Laboratory Supplies, Nottingham, UK), dehydrated through a series graded ethanol solution to water followed by incubated in 1% H_2_O_2_ diluted in Methanol for 30 min to remove blood stains. Antigen retrieval was performed in boiling 0.01 M citrate buffer for 20 min and then cooled to room temperature. Slides were blocked for non-specific binding by adding 15% goat serum (ab7481-10mL, Abcam, Cambridge, UK) diluted in 1× Tris-buffered saline with 0.1% tween-20 (TBS-T) followed by incubation in a moist chamber for 30 min. After washing, primary rabbit polyclonal anti-GFP (Abcam, Cambridge, UK ab290 1:1000), diluted in 10% goat serum, was added to sections and incubated overnight at 4 °C. Following 3× washing with TBS-T, 3,3′-Diaminobenzidine (DAB) staining was performed using Polink-2 Plus HRP Polymer and AP Polymer detection for Rb antibody kit (D39-18, Origene, Washington, USA) following manufacturer’s instructions. The slides were then dehydrated with an increasing gradient of ethanol to water followed by a final step with Histoclear. The slides were mounted with a water-free mounting medium (100579, Merk, Darmstadt, Germany) and dried overnight. Ten random images per liver sample with DAB staining were obtained using a microscope camera (DFC420, Leica Microsystems, Milton Keynes, UK) and software (Image Analysis; Leica Microsystems, Wetzlar, Germany) was utilised to capture representative images. Quantitative analysis was performed by threshold analysis using the Image J software 154 (Maryland, USA).

### 2.6. Western Blot

Liver sample lysate was prepared by adding prechilled ice-cold 400 µL RIPA buffer, with 1% Proteinase and phosphatase inhibitors (HALT) to a prechilled Lysing Matrix D tube (MP Biomedicals, 1169130-CF, Reading, UK). Then samples were lysed at 4 °C on a Precellys Evolution (Bertin Technologies, Montigny-le-Bretonneux, France) and run for 5× during 30 s cycles at 6700× *g*. Next, the lysates were centrifuged at 15,000× *g* for 10 min at 4 °C and the supernatant was collected for analysis. Membranes were probed using rabbit polyclonal anti-GFP (Abcam ab290, Cambridge, UK 1:1000). HRP-linked anti-rabbit IgG (1:200, Cell Signalling Technology, 7074, Danvers, MA, USA) was used as a secondary antibody.

Protein bands were visualised using ECL Prime Western Blotting Detection Reagent (Sigma-Aldrich, GERPN2232, Gillingham, UK) as per the manufacturer’s instructions. Quantification of the bands was performed using the iBright™ Analysis Software (Thermo Fisher Scientific, Dartford, UK). For measuring GFP expression, 30 mg of the liver was homogenised in ice-cold 1× RIPA buffer (Cell Signalling, Leiden, The Netherlands) using Precellys homogenising tube and homogeniser, centrifuged at 10,000× *g* for 20 min at 4 °C. Protein concentration was measured using BCA Protein Assay kit (23227, Thermo Fisher Scientific, Dartford, UK). For each sample, 40 µg of protein was diluted 1:1 with 2× Laemmli sample buffer (containing 10% 2-β-mercaptoethanol (β-ME)) making up 40 µL total volume, followed by vortexing and heating to 95 °C for 10 min. SDS-PAGE was used to separate the proteins at 100 V for 1 h and the wet transfer of proteins into an immobilin PVDF membrane was performed at 400 mA for 1 h. The membrane was blocked in 5% non-fat milk powder in PBS-T followed by overnight incubation at 4 °C with primary antibodies (anti-GFP; Abcam ab290 1:1000, anti-GAPDH; Abcam ab9485 1:10,000, Cambridge, UK) 3× 5 min washes with PBS-T, 1 h incubation with fluorescent secondary antibodies (IRDye^®^ 800CW Goat anti-Rabbit IgG 1:1000, 926-32210 and IRDye^®^ 680RD Donkey anti-Mouse IgG, 923-68072, LI-COR (Lincoln, NE, USA) and 3× 5 min washes with PBS-T. Image acquisition and analysis were performed using Licor Odyssey and images were analysed using LI-COR ImageStudio Lite software (Lincoln, NE, USA).

## 3. Results

### 3.1. Clodronate Administration Deplete Macrophage Infiltration in Liver and Spleen of Neonatal and Juvenile Mice

Transient macrophage depletion by systemic pre-administration of clodronate increases adenoviral-mediated liver transduction in adult mice [56]. In addition, shielding from macrophages increases liver transduction in vivo [17]. In many rare diseases, the disease severity and high unmet needs require gene therapy intervention in neonates and/or children. Therefore, we aimed to confirm that the macrophage depletion induced by systemic clodronate administration in adult mice was reproducible in both neonatal and juvenile mice. We assessed the presence of macrophages after systemic administration of clodronate liposomes in neonatal and juvenile wild-type mice (Figure 1A). Repeated systemic administration of clodronate significantly reduced the number of liver and splenic macrophages with over 2-fold reduction in both neonatal (*p* = 0.01, *p* = 0.03, respectively) (Figure 1B–E) and juvenile (*p* = 0.03, *p* = 0.04, respectively) (Figure 1F–I) C57BL/6J mice. This reduction was observed 24 h after administration, as indicated by significantly reduced expression levels of the Kupffer cell marker F4/80. Lipopolysaccharide injection used as control showed a 2-fold increase in macrophagic infiltration in the livers of neonatal and juvenile mice, and in the spleen of neonatal pups (Figure 1B–I).

### 3.2. Macrophage Depletion Enhances Lentiviral Liver Transduction in Neonatal and Juvenile Mice

We subsequently tested whether transient macrophage depletion would benefit liver transduction following in vivo lentiviral gene therapy. Here, CD1 mice received intraperitoneal injections of clodronate- or PBS-encapsulated liposomes (0.12 g/kg) at 30 and 6 h before a single intravenous injection of 4e10TU/kg of lentiviral VSV-G pseudotyped CCL.LP1.GFP vector. Untreated animals were used as negative controls. One month following gene therapy, mice were harvested and livers were collected (Figure 2A). In the group injected neonatally, liver vector genome copy number (VCN) showed a 2-fold increase in liver transduction in the clodronate- versus PBS-injected group (Figure 2B). Liver immunostaining showed a significant 10-fold increase in clodronate- versus PBS-treated animals with 22% and 2% of GFP expression, respectively (*p* = 0.02) (Figure 2C,D). The pattern of hepatocyte transduction revealed a homogeneous and scattered expression in all injected mice with no predominant expression in periportal nor pericentral hepatocytes (Figure 2D). In juvenile mice, liver VCN showed a 4-fold increasing trend of liver transduction in the clodronate- versus PBS-group (Figure 2E). These results were supported by a significant increase in GFP expression in clodronate- versus PBS-treated cohorts with 35% and 15% of heptatocytes being transduced, respectively (*p* = 0.004) (Figure 2F,G). Overall, these findings demonstrated an enhanced lentiviral-mediated liver transduction following clodronate pre-treatment in animals treated neonatally and as juvenile.

### 3.3. Macrophage Depletion Enhances Lentiviral-Mediated Liver Transduction Whilst Decreasing Splenic Transduction, a Reproducible Finding Between Outbred and Inbred Mouse Strains

To assess the reproducibility of enhanced liver transduction mediated by the lentiviral vector following clodronate pre-treatment, we replicated the experiment performed in outbred CD1 mice in inbred C57BL/6J mice (Figure 3A) [57,58,59]. CD1 mice had been initially chosen for their easy breeding, large litters, and cost-effectiveness [60,61]. We also assessed spleen VCN as an indirect marker of systemic macrophage depletion as previously published [17]. In neonates, we did not observe a VCN difference in liver (Figure 3B) and spleen (Figure 3C) between the clodronate- versus PBS-treated group. Immunostaining of the liver showed a significant (*p* = 0.04) increase in GFP production in the clodronate- versus PBS-treated group (Figure 3D,E). Compared to the PBS-treated group, the clodronate-treated juvenile animals showed a significant increase in liver VCN (*p* = 0.008) (Figure 3F), a decrease in splenic VCN (*p* = 0.0002) (Figure 3G), and an increase in GFP production in the liver (*p* = 0.006) (Figure 3H,I). The benefit of clodronate pre-treatment in lentiviral-mediated liver transduction was higher in juveniles compared to neonatal C57BL/6J mice. These results confirmed that clodronate-induced transient macrophage depletion enhances lentiviral-mediated liver transduction. Interestingly, the efficacy observed was higher in the juvenile group versus the neonatal group of inbred C57BL/6J mice. Both mouse strains achieved comparable levels of liver transduction.

### 3.4. Macrophage Depletion Does Not Influence AAV-Mediated Liver Transduction

Neonatal and juvenile CD1 mice received intraperitoneal injections of clodronate or PBS liposomes at 30 and 6 h before they received intravenous an injection of 1e13VG/kg of hepatotropic AAV8 vector (AAV8.LP1.GFP) [62]. Untreated and AAV8-only (named AAV control)-injected animals were used as negative controls. Animals were harvested at 4 weeks post-AAV administration (Figure 4A).

In neonates, liver VCN and GFP transduction or production were not increased in the clodronate- versus PBS-treated and AAV control groups (Figure 4B–D). In juvenile animals, liver VCN and GFP immunostaining did not show significant differences between clodronate- versus PBS-treated and AAV control groups (Figure 4E–G). The GFP immunostaining was <1% and 2.4% in neonates and juvenile animals, respectively. These results are consistent with AAV-mediated episomal transgene biodistribution in rapidly growing livers, with the presence of clusters of transduced hepatocytes likely associated with rare integration events. Overall, these data show no benefit of clodronate pre-treatment and transient macrophage depletion for AAV-mediated hepatocyte transduction.

### 3.5. LNP-mRNA-Mediated Liver Transduction Does Not Benefit from Macrophage Depletion

Although LNPs naturally accumulate in the liver following systemic administration [26], there is still a lack of understanding in how LNPs could interact with Kupffer cells and splenic resident macrophages resulting in potential off-target uptake. We tested the hypothesis that LNP-mRNA-mediated liver transduction might benefit from clodronate pre-treatment. Neonatal and juvenile CD1 mice were pre-treated intraperitoneally with either PBS or clodronate encapsulated liposomes 30 and 6 h before the intravenous administration of engineered LNP encapsulating GFP mRNA (Figure 5A). We previously observed that the LNP-GFP.mRNA used in this work shows early expression 30 min post systemic injection with a peak of expression at 24 h then a progressive decrease in expression lasting for some days [2]. Animals were harvested at 24 h following systemic injection. Untreated mice and mice injected with LNP-mRNA were used as negative controls only.

In neonates (Figure 5B–E) and juvenile (Figure 5E–I) mice, liver GFP western blot did not show any enhanced liver transduction. GFP immunostaining in neonatal and juvenile mice showed a significant reduction of transduction (*p* = 0.04 and *p* = 0.01, respectively). Therefore, macrophage depletion does not increase but is detrimental to liver transduction mediated by LNP-mRNA.

## 4. Discussion

In this work, we showed that clodronate-mediated transient macrophage depletion increases lentiviral-mediated liver transduction both in neonatal and juvenile mice, in both outbred and inbred strains. The decreased uptake of lentiviral vectors by macrophages mechanically increased the vector pool for on-target hepatocyte transduction. Conversely, hepatotropic AAV-mediated transduction was not modified by depletion. Clodronate pre-treatment showed a decreased liver transfection by LNP-mRNA.

The first line of defence against viral infections consists in the innate immune response induced by the complement pathway and circulating and tissue-resident macrophages. Vesicular stomatitis virus (VSV-G)-pseudotyped lentiviral vectors are opsonised by complement-mediated inactivation in human serum. This is likely due to cross-reacting, not neutralising, and complement-fixing anti-VSV-G antibodies in humans [63,64,65]. Lentiviral-binding antibodies and complement proteins can opsonise lentiviral vector particles for phagocytosis by liver and spleen macrophages and professional antigen-presenting cells [66,67,68]. As part of the immune response and complement activation upon systemic administration, high amounts of lentiviral vector transduce liver and splenic macrophages instead of hepatocytes. Indeed, following intravenous injection, lentiviral vectors will preferentially transduce Kupffer cells and tissue-resident macrophages before hepatocytes [69]. High uptake of lentiviral vector by macrophages has previously been reported in the liver and spleen, with over 70% of lentiviral DNA integrated in non-parenchymal cells in 8-week-old injected C57BL/6 mice. Fifty percent of the lentiviral vector is uptaken by the spleen in non-human primates [17]. The preferential gene transfer to the spleen by the VSV-G pseudotyped lentiviral vector, despite systemic administration and the well-described VSV-G pan tropism, may be due to the abundant blood supply to such a filtering organ [70]. Following preferential uptake by macrophages of lentiviral, a rapid inflammatory response is observed after vector administration [39,41,71,72,73].

Therefore, avoiding macrophage uptake is an appealing strategy to increase the vector pool available for hepatocytes. This strategy was successfully tested in vivo by overexpressing the “don’t eat me” CD47 antigen signal at the capsid surface of lentiviral vectors [17]. In this study, we depleted transient macrophages by using an inhibitor, clodronate, which successfully benefited liver transduction mediated by a lentiviral vector. Free clodronate has a short half-life and is rapidly cleared from the circulation by the kidney. Clodronate, a molecule from the bisphosphonate family, is routinely prescribed in clinical settings to prevent bone resorption in cancer [74,75,76]. Clodronate is generally well tolerated, with few adverse effects such as gastrointestinal disturbances, transient increase in serum creatinine, and parathyroid hormone levels [74]. Clodronate-induced macrophage depletion decreases the release of pro-inflammatory cytokines such as endotoxin-induced tumour necrosis factor alpha (TNF-α), interleukin-1 beta (IL-1β), Il-6, IL-10, and transforming growth factor beta (TGFB), which could alleviate the innate immune response against gene therapy vectors [77,78,79]. This short-term and selective depletion of macrophages has shown benefits such as increased adenoviral-mediated liver transduction and reduced humoral immune response against the transgenic protein in preclinical models [56]. Like lentiviral vectors, adenoviral vectors activate strong innate immune response through both Toll-like receptor (TLR)-dependent and independent pathways, resulting in upregulation of type I Interferons (IFNs) and inflammatory cytokines [80,81,82]. Adenoviral vectors activate the complement-mediated innate immune response via antibodies in individuals having pre-existing immunity [83].

For doses aimed at liver-targeting, AAV vectors induce a mild although detectable innate immune response, which is however to a lesser extent than the ones triggered by adenoviral and lentiviral vectors [84,85,86,87,88]. The innate immune response against AAV vectors is largely mediated by proinflammatory cytokines and chemokines in the transduced tissue due to TLR engagement but is limited and highly transient [89,90]. These molecules in turn promote immune cell induction and activation allowing the initiation and expansion of anti-transgene and/or anti-capsid adaptive immune cells, primarily CD8+ T cells [91,92,93]. “The anti-AAV innate immune response is mainly mediated by plasmacytoid dendritic cells via the TLR9 > MyD88 pathway, leading to the transition to adaptive CD8+ T cell response [94]. In contrast, macrophages and conventional dendritic cells have a limited role in sensing of AAV [89]. Cytokine profiling following AAV administration to the liver has been well-studied and has shown that multiple cytokines and chemokines are expressed, and infiltration of immune cells is observed in the first 6 h following AAV infusion. These responses are limited and rapidly exhausted within 24 h [88,95]”.

In line with our findings in neonatal and juvenile animals, the absence of AAV-mediated enhanced liver transduction following clodronate-induced macrophage depletion was previously reported in adult C57BL6/J mice [53]. The different observations between lentiviral- and AAV-mediated liver transduction following macrophage depletion by clodronate could be explained by the different innate immune responses triggered by each capsid.

LNPs have a high affinity for hepatocytes due to a rapid hepatocyte uptake mediated by the apolipoprotein E (ApoE)—low-density lipoprotein receptor (LDLr) interaction [96]. Macrophages are likely to play a limited role in LNP uptake or clearance, although macrophage depletion could theoretically still provide an incremental benefit. Varying LNP compositions, such as amino lipids, can facilitate different cell tropisms within the liver microenvironment [97]. Modifying the cholesterol structure can increase delivery to the hepatic endothelial and Kupffer cells at doses as low as 0.05 mg/kg [98]. We observed a reduced liver transfection following clodronate pre-treatment, for which the explanation is unclear.

In conclusion, our study shows that clodronate-induced macrophage depletion enhances lentiviral-mediated liver transduction in vivo. Macrophage depletion has no effect on hepatotropic AAV and decreases LNP-mRNA-mediated liver-targeting gene therapy. These findings have direct translational benefits for in vivo lentiviral gene therapy to achieve a minimal effective dose and improve safety. 

## Figures and Tables

**Figure 1 cells-13-01979-f001:**
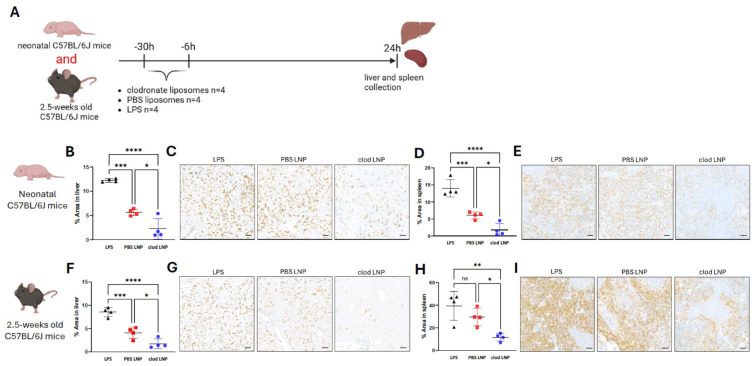
Macrophage depletion in neonatal juvenile mice in liver and spleen following clodronate liposome administration. (**A**) Schematic representation of the experimental design testing pre-treatment with clodronate liposomes, LPS and PBS liposomes in the depletion of liver macrophages in neonatal and juvenile C57BL/6J mice. (**B**) Quantification of F8/40 immunostaining, (**C**) representative images of F8/40 immunostaining in liver sections of neonatal C57BL/6J mice. (**D**) Quantification of F8/40 immunostaining, (**E**) representative images of F8/40 immunostaining in spleen sections of neonatal C57BL/6J mice. (**F**) Quantification of F8/40 immunostaining, (**G**) representative images of F8/40 immunostaining in liver sections of juvenile C57BL/6J mice. (**H**) Quantification of F8/40 immunostaining, (**I**) representative images of F8/40 immunostaining in spleen sections of juvenile C57BL/6J mice. (**B**,**D**,**F**,**H**) Horizontal lines display the mean ± standard deviation. One-way ANOVA with Tukey’s multiple comparisons test, ns: not significant, * *p* < 0.05, ** *p* < 0.01, *** *p* < 0.001, **** *p* < 0.0001; LPS (*n* = 4), PBS LNP (*n* = 4), clod LNP (*n* = 4). (**C**,**E**,**G**,**I**) Scale bars are 100 μm for ×10 magnification. Clod: clodronate liposomes; LPS: lipopolysaccharide; PBS: Phosphate Buffer Solution.

**Figure 2 cells-13-01979-f002:**
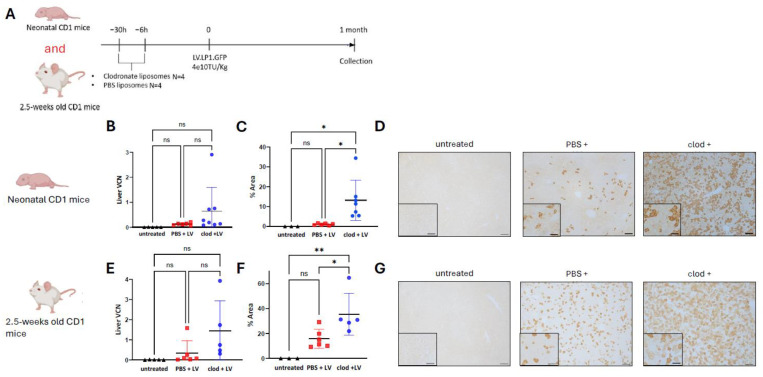
Macrophage depletion enhances lentiviral liver transduction in neonatal and juvenile mice. (**A**) Schematic representation of the experimental design testing lentiviral vector transduction following pre-treatment with clodronate liposomes in CD1 mice. (**B**) Lentiviral vector genome copies per cell in liver, (**C**) quantification of GFP immunostaining, (**D**) representative images of GFP immunostaining in liver sections of neonatally injected CD1 mice. (**E**) Lentiviral vector genome copies per cell in liver, (**F**) quantification of GFP immunostaining, (**G**) representative images of GFP immunostaining in liver sections of 2.5-week-old injected CD1 mice. (**B**,**C**,**E**,**F**) Horizontal lines display the mean ± standard deviation. One-way ANOVA with Tukey’s multiple comparisons tests, ns: not significant, * *p* < 0.05, ** *p* < 0.01; untreated (*n* = 3–5), PBS + LV (*n* = 6), clod + LV (*n* = 5–7). (**D**,**G**) Scale bars are 100 μm and 50 μm for ×10 and ×20 magnification, respectively. Clod: clodronate liposomes; LV: lentivirus; PBS: Phosphate Buffer Solution; VCN: vector copy number.

**Figure 3 cells-13-01979-f003:**
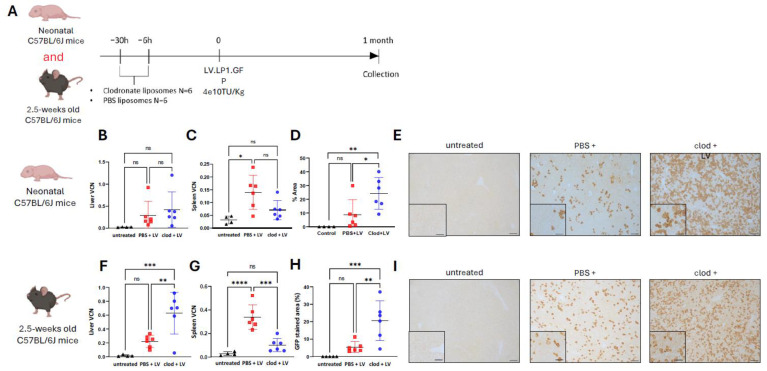
Macrophage depletion decreases splenic transduction and enhances lentiviral-mediated liver transduction; a reproducible finding between outbred and inbred mouse strains. (**A**) Schematic representation of the experimental design testing lentiviral vector transduction following pre-treatment with clodronate liposomes in C57BL/6J mice. (**B**) Lentiviral vector genome copies per cell in liver, (**C**) vector genome copies per cell in spleen; (**D**) quantification of GFP immunostaining, (**E**) representative images of GFP immunostaining in liver sections of neonatally injected C57BL/6J mice. (**F**) Lentiviral vector genome copies per cell in liver, (**G**) vector genome copies per cell in spleen; (**H**) quantification of GFP immunostaining, (**I**) representative images of GFP immunostaining in liver sections of 2.5-week-old injected C57BL/6J mice. (**B**–**H**) Horizontal lines display the mean ± standard deviation. One-way ANOVA with Tukey’s multiple comparisons test, ns: not significant, * *p* < 0.05, ** *p* < 0.01, *** *p* < 0.001, **** *p* < 0.0001; untreated (*n* = 4–5), PBS + LV (*n* = 6), clod + LV (*n* = 6). (**E**,**I**) Scale bars are 100 μm and 50 μm for ×10 and ×20 magnification, respectively. Clod: clodronate liposomes; LV: lentivirus; PBS: Phosphate Buffer Solution; VCN: vector copy number.

**Figure 4 cells-13-01979-f004:**
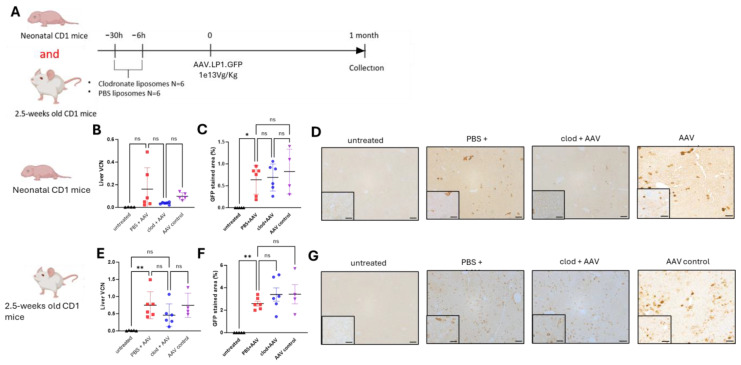
Macrophage depletion does not influence AAV-mediated liver transduction. (**A**) Schematic representation of the experimental design testing AAV vector transduction following pre-treatment with clodronate liposomes in CD1 mice. (**B**) AAV vector genome copies per cell in liver, (**C**) quantification of GFP immunostaining, (**D**) representative images of GFP immunostaining in liver sections of neonatally injected CD1 mice. (**E**) AAV vector genome copies per cell in liver, (**F**) quantification of GFP immunostaining, (**G**) representative images of GFP immunostaining in liver sections of 2.5-week-old injected CD1 mice. (**B**,**C**,**E**,**F**) Horizontal lines display the mean ± standard deviation. One-way ANOVA with Tukey’s multiple comparison test, ns: not significant, * *p* < 0.05, ** *p* < 0.01; untreated (*n* = 4–5), PBS + AAV (*n* = 6), clod + AAV (*n* = 6), AAV control (*n* = 4). (**D**,**G**) Scale bars are 100 μm and 50 μm for ×10 and ×20 magnification, respectively. AAV: adeno-associated virus, clod: clodronate liposomes; PBS: Phosphate Buffer Solution; VCN: vector copy number.

**Figure 5 cells-13-01979-f005:**
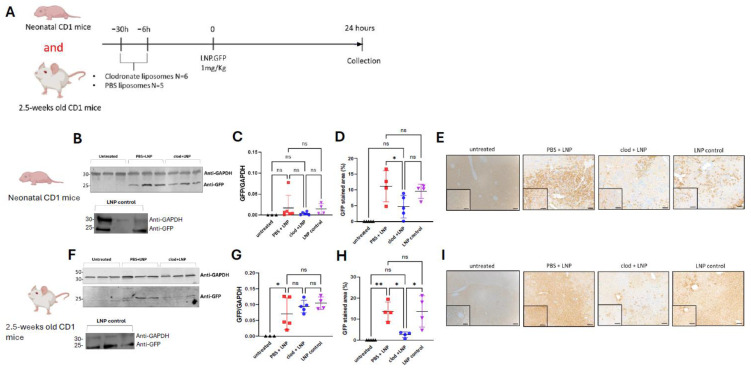
LNP-mRNA-mediated liver transduction does not benefit from macrophage depletion. (**A**) Schematic representation of the experimental design testing liver uptake of LNP.GFP following pre-treatment with clodronate liposomes in CD1 mice. (**B**) GFP western blot at 24 h post-LNP-mRNA administration (*n* = 3), (**C**) quantification of GFP western blot of livers against housekeeping control GAPDH. (**D**) quantification of GFP immunostaining, (**E**) representative images of GFP immunostaining in liver sections of neonatally injected CD1 mice. (**F**) GFP western blot at 24 h post-LNP-mRNA administration (*n* = 3), (**G**) quantification of GFP western blot of livers against housekeeping control GAPDH. (**H**) Quantification of GFP immunostaining, (**I**) representative images of GFP immunostaining in liver sections of juvenile-injected CD1 mice. (**C**,**D**,**G**,**H**) Horizontal lines display the mean ± standard deviation. One-way ANOVA with Tukey’s multiple comparisons test, ns: not significant, * *p* < 0.05, ** *p* < 0.01; untreated (*n* = 3–5), PBS + LNP (*n* = 5–6), clod + LNP (*n* = 6), LNP control (*n* = 4). (**E**,**I**) Scale bars are 100 μm and 50 μm for ×10 and ×20 magnification, respectively. Clod: clodronate liposomes; LNP: Lipid nanoparticles; PBS: Phosphate Buffer Solution.

## Data Availability

Raw data is available upon request.

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
