# Peer review of "Macrophage Inhibitor Clodronate Enhances Liver Transduction of Lentiviral but Not Adeno-Associated Viral Vectors or mRNA Lipid Nanoparticles in Neonatal and Juvenile Mice"

_cells, 2024, doi:10.3390/cells13231979_

Round 1
Reviewer 1 Report
Comments and Suggestions for Authors
This study investigates the impact of macrophage depletion on the biodistribution of AAV, lentiviral, and LNP-mRNA gene therapies in neonatal and juvenile mice. The results show that macrophage depletion increased lentiviral transduction of the liver, had no effect on AAV8 transduction, and reduced LNP-mRNA transfection. The work is very focused, the experiments are well designed, and the manuscript is well written. However, we do not agree with some of the conclusions in the discussion, and the depth of experiments is limited.
As an example, the authors often mention innate immunity and immune responses, but don’t measure any aspect of this. More relevant though, is the lack of distinction between gene transfer into Kupffer cells or hepatocytes, and the lack of granularity in the interpretation of the results. The entire effects the authors describe here could be explained by the “sink” function of macrophages and differences of the transgene. VSVg-LVs are efficiently sequestered by macrophages (which still partially register as VCN+ cells), but due to the transcriptional targeted LP1-promoter the transgene is poorly expressed in macrophages, and only the fraction of LV which transduce hepatocytes become GFP+. Consequently, clodronate treatment eliminates the sink and improves GFP+ cells (primarily hepatocytes).
For the liver tropic AAV8 (also with LP1 promoter), presence or absence of macrophages is irrelevant, since it preferentially targets liver cells in either setting.
In contrast, LNP mediated mRNA delivery lacks specific tropism and lacks any transcriptional targeting. The strong drop in GFP+ in the clodronate setting can simply be explained by the assumption that LNP preferentially transduce macrophages, not hepatocytes. The authors provide a citation (#90) to claim that macrophages play an limited role in LNP uptake, but it unclear if the citation used similar chemistry as the LNP used in the current study. Providing own experimental data could explain the observations.
Further improvements of the study are needed to address these gaps and optimize gene therapy strategies.
Major points:
1. Separating gene transfer into macrophages vs hepatocytes is essential to interpret the observations correctly. This could be done by immunofluorescence staining of the sections, or by flow cytometry.
2. The authors used immunostaining to quantify macrophage and GFP expression throughout the paper. It is recommended to also use flow cytometry for these measurements to obtain % transduced cells in addition to the %Area information.
Minor points:
1. Line 112: Maybe it would be helpful to the uninitiated reader to add that this is a lentiviral vector. Name of the vector in text and Figure2 should be identical (CCL vs LV).
2. Please add to paragraph 2.2 that it is VSVg pseudotyped LV.
3. Lines 147, 255, 260 and 268 contain typos.
Author Response
Reviewer 1
This study investigates the impact of macrophage depletion on the biodistribution of AAV, lentiviral, and LNP-mRNA gene therapies in neonatal and juvenile mice. The results show that macrophage depletion increased lentiviral transduction of the liver, had no effect on AAV8 transduction, and reduced LNP-mRNA transfection. The work is very focused, the experiments are well designed, and the manuscript is well written. However, we do not agree with some of the conclusions in the discussion, and the depth of experiments is limited.
As an example, the authors often mention innate immunity and immune responses, but don’t measure any aspect of this. More relevant though, is the lack of distinction between gene transfer into Kupffer cells or hepatocytes, and the lack of granularity in the interpretation of the results. The entire effects the authors describe here could be explained by the “sink” function of macrophages and differences of the transgene. VSVg-LVs are efficiently sequestered by macrophages (which still partially register as VCN+ cells), but due to the transcriptional targeted LP1-promoter the transgene is poorly expressed in macrophages, and only the fraction of LV which transduce hepatocytes become GFP+. Consequently, clodronate treatment eliminates the sink and improves GFP+ cells (primarily hepatocytes).
For the liver tropic AAV8 (also with LP1 promoter), presence or absence of macrophages is irrelevant, since it preferentially targets liver cells in either setting.
In contrast, LNP mediated mRNA delivery lacks specific tropism and lacks any transcriptional targeting. The strong drop in GFP+ in the clodronate setting can simply be explained by the assumption that LNP preferentially transduce macrophages, not hepatocytes. The authors provide a citation (#90) to claim that macrophages play an limited role in LNP uptake, but it unclear if the citation used similar chemistry as the LNP used in the current study. Providing own experimental data could explain the observations.
Further improvements of the study are needed to address these gaps and optimize gene therapy strategies.
We thank the reviewer for their encouraging comments.
Major points:
- Separating gene transfer into macrophages vs hepatocytes is essential to interpret the observations correctly. This could be done by immunofluorescence staining of the sections, or by flow cytometry.
We thank the reviewer for their comment. The aim of the manuscript is to increase the efficacy of in vivo lentiviral gene therapy and amplify the transgenic expression in hepatocytes. Clodronate administration does not fully deplete macrophages from data provided in Figure 1 and some macrophages will still be transduced, although affecting a much more limited number of cells. Assessing the limited remaining transduction of macrophages is out of the remit of our manuscript, which is focussing on hepatocyte transduction and ability to translate gene therapy technologies into patients.
Additionally, the transgenic expression driven by the hepatocyte-specific LP1 promoter will minimise the expression in macrophages. Therefore, immunofluorescence and flow cytometry will be of limited interest. Other methods could be used based on liver perfusion, isolation of different cell sub populations and assessing vector copy numbers (VCN). This is an experiment, which requires time and optimisation and is currently not set up in our laboratory.
- The authors used immunostaining to quantify macrophage and GFP expression throughout the paper. It is recommended to also use flow cytometry for these measurements to obtain % transduced cells in addition to the %Area information.
We thank the reviewer for their comment.
In this experimental design, we used VCN and %Area of GFP immunostaining to assess transduced cells in tissue of interest and transgenic protein expression, respectively. These are endpoints widely accepted in the field, which are used in most liver gene therapy publications. They are easy to set up and facilitate comparisons between different publications.
Flow cytometry is another approach to assess transduction efficacy, which is less popular as it requires a more complex experimental set up with liver perfusion and dissociation.
Minor points:
- Line 112: Maybe it would be helpful to the uninitiated reader to add that this is a lentiviral vector. Name of the vector in text and Figure2 should be identical (CCL vs LV).
- Please add to paragraph 2.2 that it is VSVg pseudotyped LV.
- Lines 147, 255, 260 and 268 contain typos.
Thank you for highlighting these points, which have been corrected in the revised manuscript.
Reviewer 2 Report
Comments and Suggestions for Authors
The study by Touramanidou et al. explored the impact of macrophage depletion via clodronate liposomes on liver transduction efficiency using three gene therapy vectors. The investigation revealed distinct effects of macrophage depletion on these delivery modalities, specifically enhancing lentiviral-mediated transduction, having no impact on AAV, and reducing LNP transfection. The focus on macrophage modulation introduces a unique angle for enhancing targeted delivery, with translational implications for in vivo lentiviral therapies. However, there are areas where additional explanations or experiments are needed to enhance the quality of the article.
1. Only a single post-treatment time point was assessed for AAV (4 weeks) and LNP (24 hours). Could you please explain how these time points were selected and while only single time points were chosen?
2. Regarding the production and quantification of lentiviral vectors, no details are provided regarding the integration site analysis, which is critical to rule out off-target effects or oncogenic risks.
3. Regarding the GFP expression assessment via IHC, including a quantitative analysis using flow cytometry or quantitative fluorescence imaging is suggested.
4. Regarding the fact that macrophage depletion did not enhance AAV-mediated transduction, additional characterization, such as evaluating the immune responses (by cytokine profiling) is needed.
5. The recovery dynamics of macrophages post-clodronate treatment are not discussed.
6. If possible, evaluate the immune landscape post-clodronate treatment to understand whether cytokines or chemokines influence vector transduction.
7. Also, if possible, investigate T-cell activation or neutralizing antibody formation, particularly for AAV, as these may indirectly affect transduction.
8. The finding that AAV transduction is unaffected could be linked to its relatively mild innate immune activation, which should be further discussed in discussion.
Author Response
Reviewer 2
The study by Touramanidou et al. explored the impact of macrophage depletion via clodronate liposomes on liver transduction efficiency using three gene therapy vectors. The investigation revealed distinct effects of macrophage depletion on these delivery modalities, specifically enhancing lentiviral-mediated transduction, having no impact on AAV, and reducing LNP transfection. The focus on macrophage modulation introduces a unique angle for enhancing targeted delivery, with translational implications for in vivo lentiviral therapies. However, there are areas where additional explanations or experiments are needed to enhance the quality of the article.
- Only a single post-treatment time point was assessed for AAV (4 weeks) and LNP (24 hours). Could you please explain how these time points were selected and while only single time points were chosen?
Lentiviral and AAV gene therapy injections were assessed for efficacy after a prolonged period (4 weeks) in both neonatal and juvenile mice as this timeline enables to assess long-term transduction once the liver has completed most of the liver growth at 4 weeks of age following neonatal injection.
For LNP-mRNA, the peak of transgenic protein expression is short (usually <48 hours) after injection. Therefore a 24 hours timepoint post-administration was chosen.
An Experimental design section has been added in Methods to clarify this which reads:
“Experimental design
The efficacy of intravenous infusions was assessed after 4 weeks following a single viral vector infusion for both lentiviral and AAV vectors and 24 hours after LNP.mRNA infusion. Four weeks of age represents an age when most of the mouse liver growth has been achieved [1]. The transgenic expression following LNP.mRNA is limited to few days [2] and a timeline of 24 hours was chosen to capture a peak of GFP expression.”
- Regarding the production and quantification of lentiviral vectors, no details are provided regarding the integration site analysis, which is critical to rule out off-target effects or oncogenic risks.
We thank the reviewer for their comment. Information about vector production and titration is provided in the “Vector production and formulation” section in Methods.
We agree that the risk of integration and insertional mutagenesis is key to progress towards clinical translation. However, the aim of this manuscript is to assess the potential detrimental role of macrophages in liver transduction of different modalities of viral and non-viral gene therapies and is not assessing safety of integrative vectors in vivo. We have generated data supporting safety of liver-targeting neonatal lentiviral gene therapy, including integration site analysis, which will be published in another manuscript, focussing specifically on safety of in vivo lentiviral gene therapy.
- Regarding the GFP expression assessment via IHC, including a quantitative analysis using flow cytometry or quantitative fluorescence imaging is suggested.
We thank the reviewer for this suggestion.
In this experimental design, we used VCN and %Area of GFP immunostaining to assess transduced cells in tissue of interest and transgenic protein expression respectively. These are endpoints, widely accepted in the field, which are used in most liver gene therapy publications. They are easy to set up and facilitate comparisons between different publications.
Flow cytometry is another approach to assess transduction efficacy, which is less popular as it requires a more complex experimental set up with liver perfusion and dissociation. The information provided by flow cytometry and quantitative fluorescence will look at similar endpoints than the experimental set up presented with GFP immunostaining.
- Regarding the fact that macrophage depletion did not enhance AAV-mediated transduction, additional characterization, such as evaluating the immune responses (by cytokine profiling) is needed.
We thank the reviewer for his comment. The innate immune response against AAV is essentially mediated by plasmacytoid dendritic cells via the TLR9>MyD88 pathway. This leads to the transition to adaptive CD8+ T cell response. In contrast, macrophages and conventional dendritic cells have a limited role in sensing of AAV [3]. Cytokine profiling following AAV administration to the liver has been well-studied. Multiple cytokines/chemokines are induced and infiltration of immune cells is observed in the first 6 hours following AAV infusion. These responses are limited and abolished within 24 hours [4].
As the innate sensing immune response against AAV is well characterised and has shown that macrophages play a limited role with mild and transient cytokines increase, we believe that studying cytokines after clodronate administration will add limited information to the manuscript and will not change its conclusions.
- The recovery dynamics of macrophages post-clodronate treatment are not discussed.
Thank you for highlighting this important point. Clodronate depletes macrophages within 24 hours for a period of 1-2 weeks [5].
This information has been added in the introduction section, which reads as follows (new text in italics):
“Clodronate-encapsulated liposomes achieve a transient depletion of circa 90% macrophages in both red pulp of the spleen and Kupffer cells in the liver at 24 hours after systemic injection [6, 7]. “The transient depletion of macrophages lasts for 1-2 weeks [5].”
- If possible, evaluate the immune landscape post-clodronate treatment to understand whether cytokines or chemokines influence vector transduction.
Thank you for this comment. Clodronate-induced macrophage depletion has shown consistent decrease of pro-inflammatory cytokine release (endotoxin-induced tumor necrosis factor alpha (TNF-α), interleukin-1 beta (IL-1β), Il-6, IL-10 and transforming growth factor beta (TGFB) in multiple models [8-10].
We have added this information in the discussion as follows:
“Clodronate-induced macrophage depletion decreases the release of pro-inflammatory cytokines such as endotoxin-induced tumor necrosis factor alpha (TNF-α), interleukin-1 beta (IL-1β), Il-6, IL-10 and transforming growth factor beta (TGFB), which could alleviate the innate immune response against gene therapy vectors [8-10].”
- Also, if possible, investigate T-cell activation or neutralizing antibody formation, particularly for AAV, as these may indirectly affect transduction.
Thank you for this comment. Anti-AAV immune responses can affect transduction in case of pre-existing immunity. In this work, mice had not been receiving AAV prior to the experiment and were “AAV-naïve”. We have used littermates as controls to ensure that the effect on AAV transduction could not be caused by an anti-AAV response.
- The finding that AAV transduction is unaffected could be linked to its relatively mild innate immune activation, which should be further discussed in discussion.
Thank you for this comment. We have added the following sentence in the discussion, which is reflecting the discussion about the 4th point from this reviewer:
“The anti-AAV innate immune response is mainly mediated by plasmacytoid dendritic cells via the TLR9>MyD88 pathway, leading to the transition to adaptive CD8+ T cell response [11]. In contrast, macrophages and conventional dendritic cells have a limited role in sensing of AAV [3]. Cytokine profiling following AAV administration to the liver has been well-studied and has shown that multiple cytokines and chemokines are expressed and infiltration of immune cells is observed in the first 6 hours following AAV infusion. These responses are limited and rapidly exhausted within 24 hours [4].”
References
- Cunningham, S.C., et al., Gene delivery to the juvenile mouse liver using AAV2/8 vectors. Mol Ther, 2008. 16(6): p. 1081-8.
- Gurung, S., et al., mRNA therapy corrects defective glutathione metabolism and restores ureagenesis in preclinical argininosuccinic aciduria. Sci Transl Med, 2024. 16(729): p. eadh1334.
- Zhu, J., X. Huang, and Y. Yang, The TLR9-MyD88 pathway is critical for adaptive immune responses to adeno-associated virus gene therapy vectors in mice. J Clin Invest, 2009. 119(8): p. 2388-98.
- Martino, A.T., et al., The genome of self-complementary adeno-associated viral vectors increases Toll-like receptor 9-dependent innate immune responses in the liver. Blood, 2011. 117(24): p. 6459-68.
- van Rooijen, N., N. Kors, and G. Kraal, Macrophage subset repopulation in the spleen: differential kinetics after liposome-mediated elimination. J Leukoc Biol, 1989. 45(2): p. 97-104.
- van Rooijen, N. and E. van Kesteren-Hendrikx, Clodronate liposomes: perspectives in research and therapeutics. Journal of liposome research, 2002. 12(1-2): p. 81-94.
- Van Rooijen, N. and A. Sanders, Liposome mediated depletion of macrophages: mechanism of action, preparation of liposomes and applications. J Immunol Methods, 1994. 174(1-2): p. 83-93.
- D'Souza, M.J., et al., Macrophage Depletion by Albumin Microencapsulated Clodronate: Attenuation of Cytokine Release in Macrophage-Dependent Glomerulonephritis. Drug Development and Industrial Pharmacy, 1999. 25(5): p. 591-596.
- Hu, Z., et al., Depletion of macrophages with clodronate liposomes partially attenuates renal fibrosis on AKI-CKD transition. Ren Fail, 2023. 45(1): p. 2149412.
- Song, G., et al., Liposomal sodium clodronate mitigates radiation-induced lung injury through macrophage depletion. Translational Oncology, 2024. 47: p. 102029.
- Cao, D., et al., Innate Immune Sensing of Adeno-Associated Virus Vectors. Hum Gene Ther, 2024. 35(13-14): p. 451-463.
Round 2
Reviewer 1 Report
Comments and Suggestions for Authors
The authors have addressed all comments.
Reviewer 2 Report
Comments and Suggestions for Authors
I have reviewed the revisions made to your article. The revisions are satisfactory, and the answers provided to the comments were well addressed.
I would like to express my appreciation for your timely and thorough revisions. With that, I believe the manuscript is now ready for publication.
Best regards,